# Predicting Urinary Stone Composition in Single-Use Flexible Ureteroscopic Images with a Convolutional Neural Network

**DOI:** 10.3390/medicina59081400

**Published:** 2023-07-30

**Authors:** Kyung Tak Oh, Dae Young Jun, Jae Young Choi, Dae Chul Jung, Joo Yong Lee

**Affiliations:** 1Department of Urology, Severance Hospital, Urological Science Institute, Yonsei University College of Medicine, Seoul 03722, Republic of Korea; okt1226@yuhs.ac (K.T.O.); dyjun881101@yuhs.ac (D.Y.J.); 2Department of Urology, Yeungnam University College of Medicine, Daegu 42415, Republic of Korea; urocjy@ynu.ac.kr; 3Department of Radiology, Severance Hospital, Research Institute of Radiological Science, Yonsei University College of Medicine, Seoul 03722, Republic of Korea; daechul@yuhs.ac; 4Center of Evidence Based Medicine, Institute of Convergence Science, Yonsei University, Seoul 03722, Republic of Korea

**Keywords:** artificial intelligence, neural networks, computer, ureteroscopy, urolithiasis

## Abstract

*Background and Objectives:* Analysis of urine stone composition is one of the most important factors in urolithiasis treatment. This study investigated whether a convolutional neural network (CNN) can show decent results in predicting urinary stone composition even in single-use flexible ureterorenoscopic (fURS) images with relatively low resolution. *Materials and Methods:* This study retrospectively used surgical images from fURS lithotripsy performed by a single surgeon between January 2018 and December 2021. The ureterorenoscope was a single-use flexible ureteroscope (LithoVue, Boston Scientific). Among the images taken during surgery, a single image satisfying the inclusion and exclusion criteria was selected for each stone. Cases were divided into two groups according to whether they contained any calcium oxalate (the Calcium group) or none (the Non-calcium group). From 506 total cases, 207 stone surface images were finally included in the study. In the CNN model, the transfer learning method using Resnet-18 as a pre-trained model was used, and only endoscopic digital images and stone classification data were input to achieve minimally supervised learning. *Results:* There were 175 cases in the Calcium group and 32 in the Non-calcium group. After training and validation, the model was tested using the test set, and the total accuracy was 81.8%. Recall and precision of the test results were 88.2% and 88.2% in the Calcium group and 60.0% and 60.0% in the Non-calcium group, respectively. The area under the receiver operating characteristic curve of the model, which represents its classification performance, was 0.82. *Conclusions:* Single-use flexible ureteroscopes have financial benefits but low vision quality compared with reusable digital flexible ureteroscopes. As far as we know, this is the first artificial intelligence study using single-use fURS images. It is meaningful that the CNN performed well even under these difficult conditions because these results can further expand the possibilities of its use.

## 1. Introduction

The analysis of urinary stone composition is one of the most important factors in the treatment of urolithiasis [1]. In both intraoperative and postoperative management, the composition of urinary calculi plays an important role. For example, during operations, in order to efficiently break the stone, it is important to properly select the laser energy and frequency according to the stone’s composition and size [2]. When it comes to postoperative management, depending on the stone’s components, there are various diet control management strategies, behavioral therapies, and pharmacotherapies that optimize metabolic factors and reduce the urinary supersaturation of stone composition to lower the urinary calculi recurrence rate [3].

The methods for analyzing the composition of urinary calculi are optic polarizing microscopy, scanning electron microscopy, infrared spectroscopy, X-ray powder diffraction, elementary distribution analysis, and so on. Among these methods, Fourier transform infrared spectroscopy (FTIRS) is an efficient, reliable, accurate, and rapid method and, currently, one of the most widely used [4,5]. However, it takes several weeks to receive FTIRS results, and no test can predict urinary stone composition intraoperatively or immediately after surgery.

The field of artificial intelligence is developing dramatically, and neural networks are going beyond human recognition. Neural networks demonstrate excellent performance, particularly in handling large-scale data processing, complex pattern recognition, and achieving high accuracy and consistency. Recent studies have predicted urinary stone components using neural networks and digital images. Kristian et al. reported favorable results in identifying kidney stone composition from digital photographs taken in vitro using a deep convolutional neural network (CNN) [6]. Furthermore, Estrade et al. showed decent results using intraoperative ureterorenoscopic (Olympus URF-V CCD sensor) digital images and endoscopic morphological criteria, which the authors proposed in a previous study [7,8]. Thus far, studies for autonomic recognition have utilized high-quality images, and no existing studies have used single-use flexible ureteroscopic (fURS) images. We investigated whether a deep CNN can also show decent results in predicting urinary stone composition even in single-use fURS images with relatively low resolution.

## 2. Materials and Methods

### 2.1. Study Design

This study was approved by the Institutional Review Board of Severance Hospital, Yonsei University Health System (no. 4-2022-0797). We retrospectively used surgical videos of ureterorenoscopic lithotripsy performed by a single surgeon (JYL) between January 2018 and December 2021. The ureterorenoscope used in this study was the LithoVue single-use flexible ureteroscope (Boston Scientific, Boston, MA, USA). From the photographs captured during surgery, one picture was chosen for each stone that met the pre-defined conditions. These images went through minimal image pre-processing to get rid of unnecessary blank spaces and trademarks. The results of the urinary calculi composition analysis were obtained through FTIRS and used to divide the photographs into two groups: the Calcium group and the Non-calcium group. The pre-processed images and the classified FTIRS results were used to train the CNN model.

### 2.2. Image Standardization and Pre-Processing

fURS images are affected by various factors, such as who the surgeon was and what devices were used. Therefore, image standardization is one of the key factors for decent results in this research. Each picture should include the entire surface of the stone. A single image was selected for each stone. Cases with poor visibility because of clots or debris and cases in which proper stone images could not be obtained due to video recording errors were excluded. Cases with multiple FTIRS values resulting from multi-location stones were excluded because there was no one-to-one match between the results and the stone. As a result, only cases that exactly matched the image and the stone composition analysis results were included in this study. Of the 506 total cases, 207 were finally included in this study. Regarding the bias due to differences in equipment, LithoVue has an advantage in that it has its own workstation platform. Reusable digital fURS cameras require separate workstations and light source equipment, and the choice of workstation can impact the quality of images. On the other hand, with LithoVue, it is possible to minimize the bias caused by the difference in additional equipment.

Image pre-processing was minimized in this study. In the obtained images, black margins and trademarks were deleted. Other than that, no additional processes were applied. We did not marginate the stone, mark the renal calyx, or comment at all, even if a part of the guidewire was visible in the image. The whole inclusion and exclusion process and image pre-processing are shown in Figure 1.

### 2.3. Classification of the Urinary Calculi

For each patient, the FTIRS results after surgery were collected. In this study, as a preliminary study of autonomic recognition, we tried to simplify the classification criteria given that the image quality was somewhat inferior because the images were taken retrospectively. Calcium oxalate is the most common component of urolithiasis. We hypothesized that the hardness of the stone may vary and that the cracking pattern of the stone during laser fragmentation may be different depending on the presence or absence of calcium. The endoscopic morphology classification introduced by Estrade et al. in 2021 noted a difference in morphology depending on the presence of calcium oxalate [8]. For these reasons, the FTIRS results were divided into two groups according to whether they contained any calcium oxalate (the Calcium group) or none (the Non-calcium group). There were 175 cases in the Calcium group and 32 cases in the Non-calcium group.

### 2.4. Convolutional Neural Network Model Building

CNNs were first introduced by Yann Lecun in 1989 and are now mainstream in neural network research using images [9]. Images as input data are huge, and not all areas of the data are important for classification; instead, only a specific part of the data is important, and that feature may appear anywhere in the image. Therefore, in order to use image data, a means of filtering features from huge amounts of data is required. Since CNNs extract features from image data with convolution kernels, they have an advantage in processing image data. Transfer learning is a method that uses a model that has been pre-trained and verified with high-quality data, and it can efficiently perform learning tasks with small and relatively low-quality data [10]. There are various pre-trained models. Of these, Resnet is currently one of the most widely used CNN structures. Resnet is a model that enables better network optimization through residual learning and shortcut connections [11]. In this study, we chose the transfer learning method and Resnet-18 as the pre-trained model. By applying the well-trained network from Resnet-18 to the target domain, only the new classifier layers need to be trained instead of all layers. Therefore, an advantage of transfer learning is that it can efficiently perform learning with small and relatively low-quality data. The entire CNN model training structure is shown in Figure 2.

The whole dataset was divided into a training set, a validation set, and a test set. To solve the data imbalance problem between the Calcium group and the Non-calcium group, images from the Non-calcium group were augmented to achieve the same number of images as the Calcium group. Among the 207 images, 22 were first designated as the test set, and then augmentation for the Non-calcium group was performed. Moreover, the remaining data were randomly divided into the train set and validation set in an 8:2 ratio, respectively. As a result, the training dataset included 163 images, and the validation set included 22 images. In the train and validation sets, there were 141 and 17 images from the Calcium group and 22 and 5 images from the Non-calcium group, respectively. Since the data imbalance between these two groups could distort the training process of the model, we performed three-fold data augmentation for the Calcium group and eight-fold data augmentation for the Non-calcium group. An image rotation maneuver was used for the Calcium group, and both image rotation and image flipping maneuvers were used for the Non-calcium group.

In one epoch of the training process, the model conducts model training with the train dataset and then performs an intermediate test with the validation dataset to calculate the error before propagating it back to proceed with the next training. There were seven epochs in total. Following the training, the model’s performance was finally tested with the test data. The Adam optimizer was used to optimize the model.

### 2.5. Localization Heat Maps

After building the model and completing the training, we plotted localization heat maps to analyze which part of the image had a significant influence on the decision process of the model. The gradient-weighted class activation mapping (Grad-CAM) method was used [12]. Localization heat maps were made for a total of 22 test set images. We marginated the stone in the image and quantitatively analyzed it by comparing it with the distribution of the hot spots. The images were classified into two groups depending on whether the hot spots were more or less evenly distributed within the stone.

## 3. Results

### 3.1. Stone Characteristics

When the composition of urolithiasis was classified using the Mayo Clinic classification, there were 92 (44.4%) calcium oxalate stones, 83 (40.1%) struvite stones, 28 (13.5%) uric acid stones, and 4 (1.9%) calcium apatite stones. There were no cysteine or brushite stones [13]. The Mayo Clinic classification and these results are shown in Table 1. The Korean stone composition analysis data presented in a previous study showed a distribution of 46.3% calcium oxalate stones, 29.6% struvite stones, 19.5% uric acid stones, 3.6% calcium apatite stones, 0.7% brushite stones, and 0.4% cysteine stones. Compared with these data, the proportions of calcium oxalate, struvite, and uric acid stones were similar, higher, and lower, respectively [14]. These differences may be due to the fact that this study was conducted only on cases that had undergone surgery.

### 3.2. Performance of the Neural Network Model

After the training was complete, the total accuracy in the validation set was 89.0%, and recall and precision were 86.3% and 93.6% for the Calcium group and 92.5% and 84.1% for the Non-calcium group, respectively. After training and validation, the model was tested using the test set, and the total accuracy was 81.8%. Recall and precision of the test results were 88.2% and 88.2% in the Calcium group and 60.0% and 60.0% in the Non-calcium group, respectively. The area under the ROC curve (AUC) of the model, which represents the classification performance of the model, was 0.82 (Figure 3). In general, if the AUC is 0.5 or less, the model is considered to have no discrimination capability. AUC values of 0.7–0.8 are “acceptable”, values of 0.8–0.9 are “excellent”, and values of 0.9 or higher imply “outstanding” model performance [15]. Therefore, the model used in this study can be considered to have excellent classification performance.

### 3.3. Localization Heat Maps

In the localization heat maps (Figure 4), 17 (77.3%) images had hot spots located in the stone, and five (22.7%) had hot spots outside of the stone. All five cases of hot spots outside the stone were in the Calcium group and correctly predicted by the model (true positive). Among the 18 images in the true positive group, 13 (72.2%) had a hot spot located in the stone, and five (27.8%) did not. This section may be divided by subheadings. It should provide a concise and precise description of the experimental results, their interpretation, and the experimental conclusions that can be drawn.

## 4. Discussion

The field of artificial intelligence is progressing rapidly. In the medical field, research on artificial intelligence is being actively conducted, and autonomic recognition of urolithiasis is an emerging topic. If autonomic recognition of urolithiasis is developed enough to be commercialized in the future, various aspects of the treatment guidelines for urolithiasis can be changed. For example, during ureteroscopic lithotripsy, the laser intensity can be pre-adjusted before laser firing by predicting the composition of the calculi immediately upon discovering the stone in the ureteroscopic endoscope. As a result, more efficient and faster lithotripsy may become possible. In addition, because dietary changes and behavioral therapy can be applied immediately after surgery, various stone-forming factors can be minimized. In addition, experienced surgeons can have a rough ability to predict the composition of urinary stones, but currently, there is no objective and quantitative method for such predictions. Autonomic recognition, however, allows any user, regardless of expertise, to predict the composition of urinary caculi objectively and quantitatively. Through more elaborately planned prospective studies with more high-quality data, autonomic recognition of urolithiasis can become a reality. 

The treatment methods for urolithiasis are becoming increasingly diverse and advanced. For example, endoscopic combined intrarenal surgery, which combines percutaneous nephrolithotomy with retrograde ureteroscopy, is being widely used as it shows higher stone-free rates for complex stones compared with traditional percutaneous nephrolithotomy alone [16]. Additionally, robotic stone surgery has gained attention as it reduces radiation exposure for the surgeon and assistant while achieving good treatment outcomes [1,17,18]. If autonomic recognition technologies are combined, they could lead to even faster operations and better results.

This is the first study to present a CNN model for autonomic recognition using single-use fURS images. The LithoVue single-use flexible ureteroscope has a CMOS image sensor, which is inferior in image quality and sensitivity to the CCD image sensor in the Olympus URF-V [19]. However, it has several advantages. First, it is cost-effective. Reusable digital flexible ureteroscopes cost more to purchase, repair, service, clean, and sterilize. By contrast, single-use flexible ureteroscopes have no maintenance-related costs other than purchase and storage costs [20]. Second, they have less risk of contamination. Maintenance of reusable digital flexible ureteroscopes inevitably requires the use of high-level disinfection methods because, if not properly sterilized, they can transmit infections [20]. Since LithoVue is a single-use flexible ureteroscope that does not have this problem, it has an advantage in terms of the risk of possible contamination. Third, single-use flexible ureteroscopes have an advantage in research using medical images. It is essential for researchers to consider variables that are changed by the different protocols or machines used in each hospital. However, LithoVue has its own workstation platform, and the monitor, light source, and image processing software are all mounted on a single mobile cart. Thus, there is no need to consider mechanical differences in research using LithoVue. Single-use flexible ureteroscopes are currently used by many hospitals because state-of-the-art devices cannot be supplied to all institutions for economic reasons. It is significant that the accuracy of the CNN model can reach 86.0% even with single-use fURS images.

In this study, transfer learning was chosen as a method of CNN model building. Transfer learning is a machine learning method that uses a pre-trained model as the starting point for a new target model. A pre-trained model is one that has already been trained on a large number of high-quality images and whose performance has already been verified. Transfer learning has the advantage of being able to create a model with good classification performance even with a few low-quality images. Therefore, it can be highly recommended to consider applying for studies of low-quality images and diseases with few cases due to low incidence rates.

This study has another important implication in that it proceeded with minimal supervised learning. We only used images after minimal pre-processing and classification of the results of the urinary calculi component for model training. In this study, there was no need for the researchers to classify the morphology of the stone or to marginate any stone or other anatomical findings. As each image pixel is data in itself and the model interprets and learns patterns from the data through convolution, it is assumed that good results can be obtained even if the intervention of the researcher is minimized.

We created localization heat maps, and the hot spots were located in the stone in 17 cases (77.3%). This result serves as significant evidence that the model focused on the stone itself rather than other structures within the image, such as renal parenchyma or guidewire, to predict the composition of the stone. However, in this study, there was no case in which the hot spot was outside of the stone in the Non-calcium group, and this seems to be because the number of cases was too small.

Data imbalances are one of the most important issues in neural network research. It is ideal to have data in equal proportions for each group in machine learning research, but this balance is difficult to achieve in the real world. In this study, the Calcium group included 175 cases, and the Non-calcium group included 32 cases. To overcome the data imbalance problem, we conducted image augmentation. Image augmentation was performed by image flipping and rotation maneuvers. In this study, the data imbalance problem was solved with a relatively simple method because the data were simply classified into two groups. However, complex classification is required to enable autonomic recognition in the future, and the issue of data imbalance should be dealt with in greater detail.

This study has several limitations. First, this study was retrospectively designed. The images used in this study were inevitably of lower quality than those taken precisely in prospectively planned studies. In addition, section images of urinary calculi were not included in this study. The composition of the surface and core can differ in urolithiasis [21]. If section images are included in later research and images of better quality can be taken, more detailed predictions of composition can be possible, and the performance of the neural network model can be dramatically improved. Second, the FTIRS results were subject to only binary classification, which divided them into the Calcium and Non-calcium groups. There are many different components of urolithiasis, including uric acid, struvite, brushite, cysteine, and so on. In addition, the pathogenesis and etiology of urinary calculi formations differ by composition. To apply appropriate behavioral or dietary management strategies to patients in actual clinical practice, it is necessary to predict the detailed components. The finer the classification, the more complex artificial intelligence models are needed. With the development of artificial intelligence technology and further studies using high-quality data, it will be possible to solve this problem. Although this study has several limitations, it has significant meaning in the field of autonomic recognition of urolithiasis as it is the first study using single-use fURS images, and the CNN showed decent results even with a relatively small number of cases and low-quality images.

## 5. Conclusions

Single-use flexible ureteroscopes have financial benefits, but they have low vision quality compared with reusable digital flexible ureteroscopes. As far as we know, this is the first artificial intelligence study using single-use fURS images. It is very meaningful that the performance of CNN showed good results even under these difficult conditions, in that it can further expand the possibilities of CNN’s use. If autonomic recognition of urinary stone composition during operations becomes possible in the future, the paradigm of urolithiasis management may change.

## Figures and Tables

**Figure 1 medicina-59-01400-f001:**
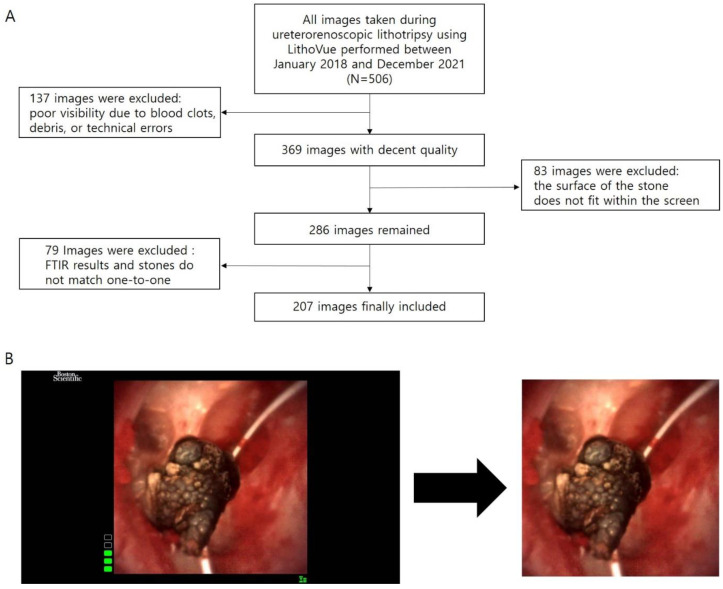
(**A**) Inclusion and exclusion processes and (**B**) image pre-processing. Image pre-processing was minimized by erasing only the black margins and leaving the central image. Note that the guidewire is still visible in the final image. FTIR: Fourier transform infrared.

**Figure 2 medicina-59-01400-f002:**
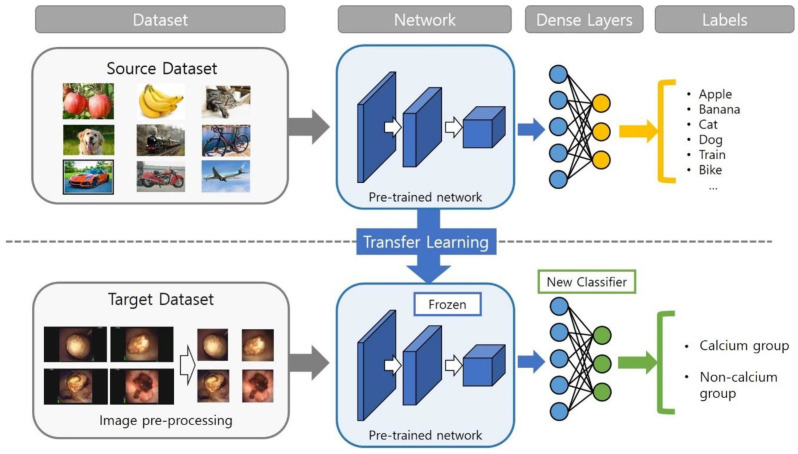
Structure of the model used in this study.

**Figure 3 medicina-59-01400-f003:**
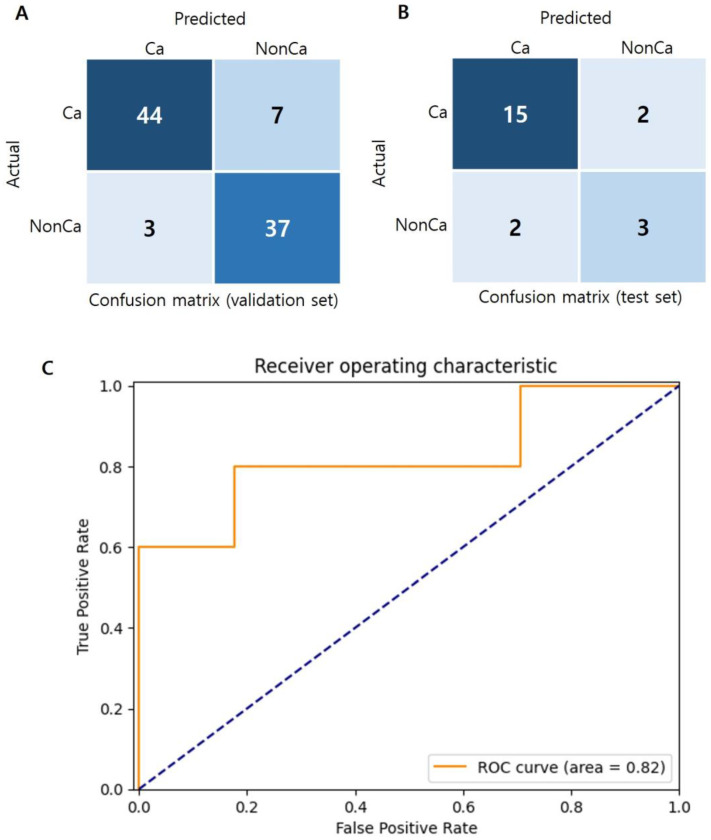
(**A**) Confusion matrix of the validation set; (**B**) confusion matrix of the test set; and (**C**) receiver operating characteristic (ROC) curve. Ca: Calcium group; NonCa: Non-calcium group.

**Figure 4 medicina-59-01400-f004:**
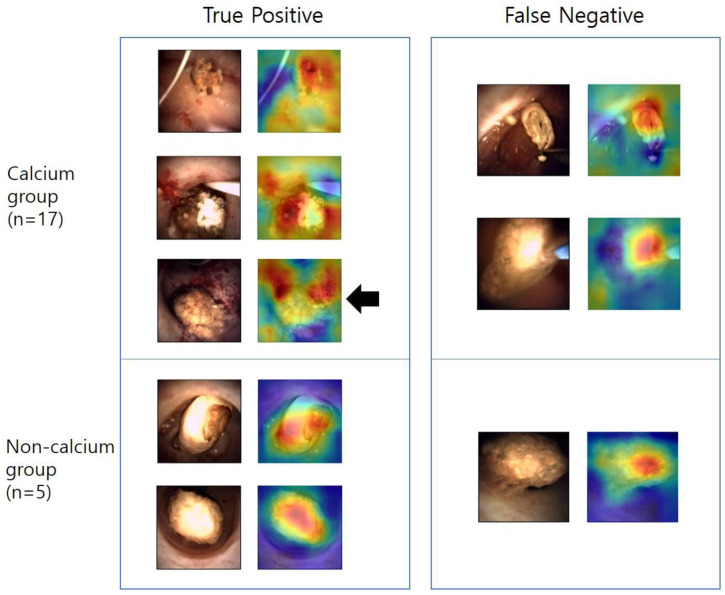
Localization heat maps. The black arrow indicates the case in which the hot spots are scattered on the outside of the stone. Of the 22 test images, 17 belonged to the Calcium group and five belonged to the Non-calcium group. There were five images with hot spots outside the stone, and all of these cases were in the Calcium group and the true positive group.

**Table 1 medicina-59-01400-t001:** The composition of urolithiasis by Mayo Clinic classification.

Stone Composition	Value
Calcium oxalate	92 (44.4%)
Struvite	83 (40.1%)
Uric acid	28 (13.5%)
Carbonate apatite	4 (1.9%)
Total	207

Data are shown as numbers (%).

## Data Availability

The datasets generated during and/or analyzed during the current study are available from the corresponding author on reasonable request.

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
