# Peer review of "Predicting Urinary Stone Composition in Single-Use Flexible Ureteroscopic Images with a Convolutional Neural Network"

_medicina, 2023, doi:10.3390/medicina59081400_

Round 1

Reviewer 1 Report

First of all, I congradulate the authors for this study. It seems that in the near future, technologies such as artificial intelligence and neural network will find more place in daily practices in urology. 

I only would recommend to correct the term of "reusable flexible ureteroscopes" to "reusable digital flexible ureteroscopes.

Author Response

Thank you for your great commment. Based on your comment, we have changed the term from 'reusable flexible ureteroscopes' to 'reusable digital flexible ureteroscopes'. We are grateful as it allows us to use a more accurate term.

Reviewer 2 Report

This study aimed to predict the stone composition (Ca stone or non-Ca stone) from the intraoperative images captured by single-use endoscopy. They use CNN, one of the machine learning algorithms, and showed an 80% accuracy. 

The study design was straightforward and easy to follow. The result was reasonable. Because automatic prediction of stone composition has been reported elsewhere, the novelty of this study is to use images from single-use endoscopy. 

For experienced surgeons, distinguishing CaOx stone from struvite stone is not so difficult.  The authors should try to discuss why AI prediction is important.

Author Response

Thank you for your great comment. We stated in the first paragraph of our discussion that with the advent of autonomic recognition, it would be possible to adjust to the optimal laser intensity before laser firing, enabling a more efficient surgery. Additionally, we highlighted the advantages of being able to initiate dietary and behavioral therapy immediately after surgery. As you mentioned in your comment, experienced surgeons are already capable of distinguishing the composition of kidney stones to some extent, and AI prediction allows for objective quantification of these predictions, benefiting both beginners and experienced surgeons. This comprehensive explanation provides a more detailed demonstration of why autonomic recognition of urinary calculi is crucial. We added this content to first paragraph of discussion part. (p.7; line 11-15)